# Design of Reliable Remobilisation Finger Implants with Geometry Elements of a Triple Periodic Minimal Surface Structure via Additive Manufacturing of Silicon Nitride

Christof Koplin [1,*], Eric Schwarzer-Fischer [2], Eveline Zschippang [2], Yannick Marian Löw [3], Martin Czekalla [3], Arthur Seibel [3], Anna Rörich [4], Joachim Georgii [4], Felix Güttler [5], Sinef Yarar-Schlickewei [6] and Andreas Kailer [1]

1  Fraunhofer-Institut für Werkstoffmechanik (IWM), Wöhlerstraße 11, 79108 Freiburg im Breisgau, Germany
2  Fraunhofer-Institut für Keramische Technologien und Systeme (IKTS), Winterbergstraße 28, 01277 Dresden, Germany
3  Fraunhofer-Einrichtung für Additive Produktionstechnologien (IAPT), Am Schleusengraben 14, 21029 Hamburg, Germany
4  Fraunhofer-Institut für Digitale Medizin (MEVIS), Max-Von-Laue-Straße 2, 28359 Bremen, Germany
5  Institut für Diagnostische und Interventionelle Radiologie, University Hospital Jena, Friedrich-Schiller-University, Am Klinikum 1, 07747 Jena, Germany
6  Klinik und Poliklinik für Unfallchirurgie und Orthopädie, Universitätsklinikum Hamburg-Eppendorf, 20251 Hamburg, Germany
*  Correspondence: christof.koplin@iwm.fraunhofer.de; Tel.: +49-76-1514-2269

**Abstract:** When finger joints become immobile due to an accident during sports or a widespread disease such as rheumatoid arthritis, customised finger joint implants are to be created. In an automated process chain, implants will be produced from ceramic or metallic materials. Artificial intelligence-supported software is used to calculate three-dimensional models of the finger bones from two-dimensional X-ray images. Then, the individual implant design is derived from the finger model and 3D printed. The 3D printing process and the structures used are evaluated via model tests and the final implant design via a reliability calculation in a way to ensure that this is also possible via an AI process in the future. Using additive manufacturing with silicon nitride-based ceramics, model specimens and implants are produced via the lithography-based ceramic vat photopolymerisation process with full geometry or elements of triple periodic minimal surfaces structure. The model specimens are tested experimentally, and the loads are matched with a characteristic strength assuming a Weibull distribution of defects in the volume to generate and match failure probabilities. Calculated fracture forces of the silicon nitride-based ceramic structure was validated by comparison of simulation and tests, and the calculation can be used as a quality index for training of artificial intelligence in the future. The proposed method for individualized finger implant design and manufacturing may allow for correction of potential malpositions of the fingers in the future.

**Keywords:** remobilisation; additive manufacturing; reliability; AI-based; reconstruction; joint-implant; crack-growth; autogeneration

## 1. Introduction

In Germany alone, at least five million people suffer from symptomatic arthrosis, and one and a half million people suffer from rheumatic diseases [1]. Up to now, the possibilities of implant treatment and remobilisation in the area of small joints have been insufficient, as the design of the implants does not allow safe functionality due to their small size and high mechanical requirements. Challenges in the field of finger endoprostheses are the restoration of the morphology, mobility, stability and loading capacity, low bone loss during implantation, a stable and durable fixation at the bone, and wear-resistant sliding contact surfaces [2]. In contrast to the classical endoprostheses for the replacement of knee and hip joints, finger bones are much thinner. There is far less musculature to support the implants, so the risk of implant loosening is higher than with large implants. Studies from the 1960s

showed increased loosening and breakouts in finger joint prostheses caused by inadequate material properties and lack of osseointegration [3]. Until today, finger joint implants are therefore uncommon and stiffening surgical techniques are widespread. This contradicts the desire of all patients with rheumatoid arthritis to be able to move their wrists again. Less than 20% of patients are pain-free and the complication rate is 40% [4–7].

Our goal is to provide patients who, due to a wide variety of diseases, can no longer move their joints in the hand and finger area, or can only do so to a very limited extent, with new types of individually adapted implants and thus restore their ability to move. The current therapeutic approach is to treat the damaged joint with medication or stiffen it by means of a so-called arthrodesis. Pain and/or restricted mobility of the joints may lead to significant restrictions in patients' quality of life as well as high follow-up costs due to increased care and therapy expenditure. Replacing this approach and enabling patients to be remobilised is a desirable goal which offers a decisive advance compared to current forms of therapy.

The research work undertaken for this purpose covers the entire process chain and concentrates on the reconstruction, development, manufacture, and clinical, standard-compliant implementation of individually manufactured ceramic implants using the finger joint as an example. A central component is the use of artificial intelligence (AI) for the autogeneration of individualised implant designs from X-ray and computer tomography (CT) images of the patient. Based on the transfer of data from 2D imaging procedures, implants are designed with osseointegrative surfaces, manufactured from customised materials using new near net shape and additive manufacturing methods, and tested mechanically and biologically in vivo. For the first time, the successful mapping of the process chain via a platform for finger joints opens the opportunity to maintain or restore the joint mobility required by the patient. The presented remobilization approach can be used for various indications such as arthrosis and arthritis as well as for traumatology.

The aim of this paper is to show the achievements and possibilities, using $Si_3N_4$ implant material for a representative example. For comparison, ATZ, Ti-alloys, and steel were also investigated in the same setting, but results are omitted.

### 1.1. Biomechanical Simulation and Imaging

Finger joints are subject to constant stress in the form of pressure loads, such as when playing the piano, to extreme tensile and shear loads on the bone, such as in sports climbing. In sports medicine [8], compressive loads of 100 to 500 N per finger can be determined, although biomechanical principles continue to be part of basic research. The experimental measurement of internal forces in joints, bones, muscles, tendons and ligaments is one of the challenging tasks of biomechanics and at the same time extremely elaborate and complicated both in vitro and in vivo [9]. To simulate the biomechanical processes and the resulting loads, the microstructure of biological tissue is homogenised computationally. Computer-aided methods in biomechanics are mainly multi-body systems (MBS) or finite element methods (FEM). In the meantime, it is possible in commercial program packages to couple FEM and MBS. Modern methods for determining the necessary boundary conditions for modelling are carried out, e.g., using motion capturing. One of the main difficulties in simulation is to model the geometry and the bone structure as correctly as possible. Usually, the shape and density distribution can be determined on the basis of 3D images from CT. Modern micro-CT images allow characterisation of the anisotropic material behaviour of the cancellous bone even on small samples. In the biomechanical simulation of implant systems, the difference in stiffness between biological and artificial material remains significant. Bones have an average modulus of elasticity of approx. 17 GPa [10], while silicon-nitride ceramic has a Young's modulus of 300 GPa, alumina-toughened-zirconia 230 GPa, and biomedical approved titanium alloys in the range of 60 to 130 GPa. This difference becomes more critical when considering osteoporotic bone structures with reduced stiffnesses and strengths. In contrast to the internationally recognised method of osteodensitometry for osteoporosis, high-resolution imaging peripheral computed tomography enables the

visualisation of bone fine structure [11]. Osteodensitometry cannot image trabecular structures, whose destruction along the bone resorption process is decisive. Measurements of the bone fine structure of spongiosa cylinders from vertebral bodies in CT images showed, for example, stiffness values of only 40 to 400 MPa and extremely reduced strength.

### 1.2. AI-Based Reconstruction of the Joint and Implant Generation

AI is already successfully used in many medical diagnostics systems [12], but further research is needed to compare AI alone and as an adjunct with human experts in implant identification [13]. Every personalised implant is based on a 3D computer model of the patient's individual anatomy, which is extracted from medical image data. In today's clinical practice, 2D X-ray images are routinely used due to the rapid availability of 2D X-ray scans (typically immediately) compared to 3D CT images (1–2 weeks) and the costs involved. The 3D anatomy can be reconstructed from such 2D images using classical optimization or AI approaches, see the overview article [14]. During 3D reconstruction, geometrical information (e.g., contours of segmented structures) and intensity information taken from the image are usually used. A common tool for representing 3D shapes are statistical shape models (SSMs), which describe 3D geometries via suitable (reduced) parameter spaces [15].

Before reconstructing a 3D anatomy, bone structures have to be segmented in 2D X-ray and 3D CT images. AI approaches using Convolutional Neural Networks (CNNs) are the current state of the art [16].

The precise fitting of orthopaedic implants can be automated with the help of neural networks. In the field of patient-specific implants, the number of studies is low due to the novelty of the research area. Statistical shape models are already being used to improve the accuracy of fit of implants—the focus here, however, is on creating a generic implant that fits as large a proportion of the population as possible [17,18]. Automated creation of individual implants is not yet possible.

### 1.3. Materials and Processes for Implant Manufacturing

Titanium alloys are very often used for biomedical applications, especially for endo-prostheses. In addition to classical manufacturing processes such as forging or casting, for example for hip implants, powder-based manufacturing technologies such as metal powder injection moulding have now also found applications in biomedicine [19]. In addition to the biocompatibility of the material, the biomechanical adaptation of the mechanical properties to the human bone is an essential criterion for implant design and material selection [20]. Both modern near net shape procedures and additive manufacturing can mimic the natural cellular structure of the bone, which makes it possible to adapt the geometry to the individual patient and at the same time achieve a mechanical adaptation of the stiffness (the modulus of elasticity) to the surrounding bone [21].

Oxide ceramics, such as $Al_2O_3$, have been successfully approved and used in medicine for many years. In endoprosthetics, the mixed ceramic Zirconia-Toughened-Alumina is used for hip, knee and shoulder endoprosthetics. In the field of implantology, Yttria stabilized zirconoxide ceramics have become established for crown and bridge frameworks. The implants are manufactured using a conventional ceramic shaping technology consisting of powder conditioning, densification, thermal treatment and hard machining. Silicon nitride was found to show a decreased bacteria activity on the surface [22] and to be an ideal ceramic for medical applications [23]. A broader overview of its benefits was already given by [24]. Due to the excellent mechanical properties of silicon nitride, investigations of tribological behaviour are being extended. The main activities in the field of silicon nitride for implants are currently in the spine area and are being pushed by a US company [25].

According to Jariwala et al. additive manufacturing technologies offer new opportunities for bone tissue engineering. After bone trauma, tumor, infection, non-occlusive fracture, or congenital anomaly, regeneration of the form and function is clinically important. Redesigning the anatomical form and structure of bone tissue has seemed impossible until

now because the structure and properties of bone are individual and damage is equally unique [26].

Triply periodic minimal surfaces (TPMS) can be described by a non-intersecting 3D surface with a mean curvature equaling zero [27]. According to the review by Feng et al. [28] the corner- and edge-free surfaces of TPMS are particularly suitable for cell attachment and growth. A high adjustable surface area with large cavities at a large number of connection sites are ideal for flow function and exchange reactions. Therefore, TPMS structures are already widely used in biotechnology. The properties under mechanical uniaxial compression have already been studied for a large number of materials. The use of heterogeneous TPMS and or even multiscale TPMS are still under research according to current knowledge. Ceramic based TPMS structures have been built on $ZrO_2$ [29], HA [30], SiC [31], and others, but $Si_3N_4$ still has yet to be attempted.

The combination of $Si_3N_4$ ceramics, a circular gradient structure for finger implants, and reliability calculation will be demonstrated in this paper.

## 2. Materials and Methods

### 2.1. 3D Model Generation from 2D Medical Images

For model construction and validation, 81 3D CT scans and pairs of 2D X-ray images from 200 hands from University Hospital Jena and 126 3D CT scans (of which 40 patients have corresponding 2D X-ray images) from University Hospital Hamburg Eppendorf were studied. nnU-Nets were trained to segment the finger bones (metacarpal, proximal, intermediate, distal) in 2D and 3D [32]. Reference segmentations for the training were done manually on a subset of the images using the SATORI annotation platform [33]. The nnU-net segmentations on the remaining images were reviewed by a radiology technician and corrected if necessary. Finally, surface meshes of the finger bones were generated from these segmentations.

The SSM construction pipeline starts with a rigid prealignment of all bone segmentations to a common reference system. Thereafter, volumetric deformable image registration, followed by non-rigid surface registration, is performed separately for each finger bone [34]. The actual SSMs for each finger bone are then computed from Procrustes Analysis and principal component analysis. Surface meshes of incomplete, i.e., broken or cut, structures were discarded, so the actual number of instances used for each SSM varies. The presented 2D–3D matching algorithm relies on SSMs constructed from approximately 184 different finger bone surface meshes. The resulting SSMs describe the variance in the geometry of each finger bone using about 18 shape parameters, called modes. All image processing tasks were done in MeVisLab [35].

We follow a contour-based approach for reconstructing 3D models from 2D X-ray images. Here, the idea is to minimize the distance between the finger bone contours extracted from the segmentations of the 2D X-ray images and the projection of the corresponding SSM. For reaching a minimal distance, the orientation and position of the SSM in space and its modes can be varied. Optimization is done using a bounded version of the Nelder-Mead simplex (direct search) method implemented in Matlab [36] using the fminsearchbnd routine [37]. In a first step, the transformation parameters are preoptimized while setting all modes to zero. Starting from the found transformation parameters and zero modes, a second optimization step is used for fine tuning of the transformation parameters and optimization of the modes. Modes can be regularized by applying different norms. The 3D model reconstruction pipeline is visualized in Figure 1.

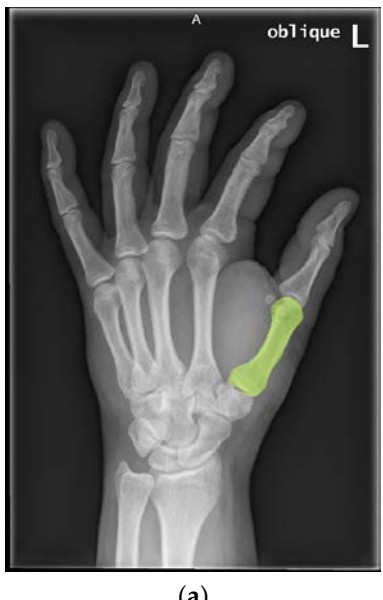
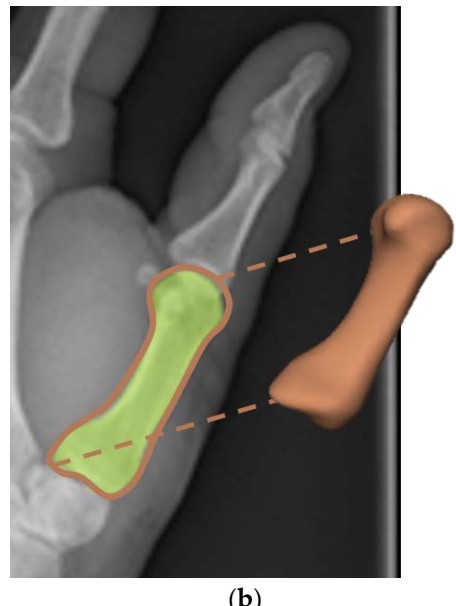

(**a**)                                        (**b**)

**Figure 1.** 3D model reconstruction pipeline: (**a**) original X-ray with automatically segmented 2D thumb metacarpal bone (green overlay); (**b**) 3D reconstructed shape (orange) generated by fitting the shape model projection (orange contour) to the 2D segmentation.

A validation data set was constructed from random but known modes and prescribed translation and rotation applied to the SSMs. For each finger bone, five translation-rotation configurations were tested: no rotation + no translation, 30° rotation around *x*-axis + prescribed translation, 30° rotation around *y*-axis + prescribed translation, 30° rotation around *z*-axis + prescribed translation, and 30° rotation around all axes + prescribed translation. The outlines of the modified SSMs projected onto the x-y-, x-z-, and y-z-plane were treated as artificial segmentation results and used as input for the optimization process.

*2.2. Design Process*

The software nTopology [38] was used to create the implant geometries. Models of the reconstructed joint bones of the middle hand and fingers were imported as mesh (STL) files and converted into an implicit model geometry for further processing. Manually selected landmarks on the implicit bone geometries were used to determine the alignment of the joints. The actual joint implant was created based on the bone geometry; see Figure 2. The head of the implant that will later be integrated into the middle hand bone has a convex, longitudinal oval shape (condyle), and the counterpart in the adjacent finger is equipped with an inwardly (concave) curved groove that matches the implant of the middle hand bone. The articular surfaces were laterally coupled through the form-fit of the two condyles. There is no axial coupling, which should typically be restored by reconstructing the ligamentous structure.

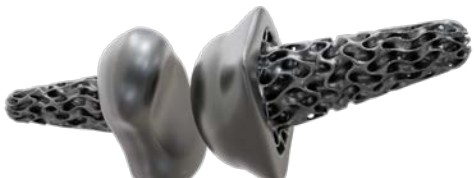

**Figure 2.** CAD model of the finger joint implant.

*2.3. CerAMfacturing of Test Components*

$Si_3N_4$ is particularly suitable for being used as material of patient specific ceramic implants [39–41], since it offers a suitable surface chemistry and high mechanical properties.

Therefore, a Si$_3$N$_4$ composition suitable for biomedical applications was used for investigations and the development of a new photoreactive suspension that can be used within a lithography-based additive manufacturing (AM) process, especially with the CerAM VPP technology. Lithography-based Ceramic Manufacturing (LCM, Lithoz GmbH, 1060 Wien, Austria) is one example of this technology and used for the manufacturing of all test components presented in this study. A detailed description of the process can be found in [42] and the development of the Si$_3$N$_4$ suspension is described in [24].

All test samples—discs (green body size: diameter d~49 mm, height h~2.6 mm), TMPS and a test-implant with a gradient SplitP TPMS—were manufactured via CerAM VPP by using the developed Si$_3$N$_4$ suspension of [24] with a solid content of approx. 40 vol%.

After manufacturing the ceramic green components, cleaning of residual suspension was done with a cleaning solution (Lithasol30, Lithoz GmbH, 1060 Wien, Austria) in combination with compressed air. Afterwards the cleaned green components were visually inspected concerning quality and the outer dimensions in the three spatial directions (x, y, z, related to the manufacturing orientation) were determined with a caliper. According to the usual ceramic processing, the green components were debindered and sintered as described in [24].

### 2.4. Mechanical Strength

The mechanical strength of the additively manufactured Si3N4 (VPP Si$_3$N$_4$) was determined via ball-on-three-balls (B3B) and compression tests.

Discs with a thickness of 1.5 mm and a diameter 36 mm were machined in order to determine the inert strength $\sigma_c$, characteristic strength $\sigma_0$, and the Weibull modulus m. Danzer et al. and Nohut et al. [43,44] described the limitations and the application of the B3B tests based on ISO 14704. A total of 10 Constant stress-rate (CSR) tests were executed at 20 °C (denoted by RT) at a load rate of 100 N/s. Cyclic fatigue tests were conducted in phosphate buffered solution at 37 °C up to 5 times $10^6$ cycles based on ISO 13356 with a frequency of 15 Hz and an R-ratio of 0.1. A total of 14 specimens were tested using a maximal stress of the cyclic load at 80% (5 specimens), 70% (4 specimens) and 60% (5 specimens) of the characteristic strength $\sigma_0$. The geometry of the B3B setup is defined by 4 ceramic spheres with radius 12mm and 3 spheres positioned at the backside 15 mm from the center of the 4th sphere and the center of the disc. A biaxial tensile stress is formed at the centre. The calculation of the stress can be done using a geometric function f, the applied force F, and the thickness of the disc.

$$\sigma_{max} = f\frac{F}{t^2} \tag{1}$$

The geometrical function f depends on the radii of the plate, the center position of the balls, the thickness t, and the poisson constant. Börger and Staudacher et al. offer a full analysis of the function and geometrical limits [43,45–47].

For conducting a uniaxial compression test, a self-built electromechanical central spindle testing machine with an Instron 8500plus control system (Instron GmbH, Darmstadt, 64293 Darmstadt, Germany) was chosen. The even plates in contact with the specimen were made of aluminium oxide. After aligning of the loading axes, the load was applied via a centered spherical contact to prevent any nonparallel load introduction. The tests were done in force control at 100 N/s in air at room temperature. A specimen of 15 mm uniaxial height and a diameter of 10.5 mm were loaded until fracture.

Results of 4-point bending tests were used for comparison. The 4-point bend bars were chosen with dimensions of 3 × 4 × 45 mm$^3$ and machined according to DIN EN 843-1 [48]. The tests were also conducted according to DIN EN 843-1. Constant stress-rate (CSR) tests were performed on 4-point bend bars at 20 °C at load rates of 200 N/s. Cyclic fatigue tests were carried out in lab air at 20 °C. For each combination of load rate and temperature 30 samples were characterized as in [49].

The finite-element-simulations were done using the abaqus product suite of dassault systems [50]. CAE was used for meshing and simulation setup. Standard was used for stress-displacement analysis and viewer for data processing. The subroutine user defined field was used for calculation of survival probabilities based on element volume and stress states using the Weibull parameter. Abaqus has a built-in python-based scripting interface that is suitable for future integration into AI loops.

For brittle ceramics, the calculation of a failure probability via defects distributed in the volume according to a Weibull distribution is assumed. Over the entire component volume ($V$), the failure probability of an applied stress distribution under static load is:

$$f = 1 - exp\left(-\int_{V_{comp}} \frac{1}{V_0}\left(\frac{\sigma(x,y,z)}{\sigma_0}\right)^m dV\right). \tag{2}$$

By defining the stress distribution in the component such that:

$$\sigma(x,y,z) = \sigma_{c,\,comp} \cdot f(x,y,z), \tag{3}$$

where $f(x, y, z)$ is a distribution function, and $\sigma$ is a load-dependent stress quantity at the location (for ceramics, this value is equal to the maximum principal stress). In this way, one can define an effective volume of the component ($V_{eff,comp}$), and this is given analytically or by finite element representation as:

$$V_{eff,\,comp} = \int_{V_{comp}} f(x,y,z)dV, \tag{4}$$

$$V_{eff,\,comp} = \sum_i f(x_i,y_i,z_i)V_i \tag{5}$$

The Weibull material scaling parameter ($\sigma_0$) can now be calculated using the effective volume of a component ($V_{eff,comp}$), by means of the scaling volume ($V_0$). These parameters can be estimated from the strength results of CSR-tests and by characteristic Weibull strength from the tests (comp: B3B and 4PB):

$$\sigma_0 = \left(\frac{V_{eff,\,comp}}{V_0}\right)^{1/m}\sigma_{c,comp}. \tag{6}$$

If a high linear correlation is found in a Weibull-plot of the individual strength results, the description by a distribution of critical defects is applicable. The slope of the linear correlation is the Weibull parameter m. To estimate the effective volume, the loading of a component was done by finite element simulation and numerical integration via a subroutine.

($da/dN$) is used to describe cyclic crack-growth rate for brittle material such as ceramics. The number of load cycles $N$, the growth rate by a Paris-law expression in terms of the applied stress intensity range $\Delta K_I$, and the fracture toughness $K_{IC}$, are needed in addition of two parameters, namely, a material parameter $A$ and the crack-growth exponent $n$ [51]. For a description:

$$\frac{da}{dN} = A(\Delta K_I)^n = A^*\left(\frac{\Delta K_I}{K_{Ic}}\right)^n \tag{7}$$

Subsequently the stress range $\Delta\sigma$ can be given in terms of the number of cycles to failure ($N_f$) by:

$$log\Delta\sigma = \frac{1}{n}log\left(B\sigma_c^{n-2}\right) - \frac{1}{n}log\left(N_f\right) \tag{8}$$

Here, $-1/n$ is the slope of the $log\Delta\sigma$ vs. $logN_f$ plot.

## 3. Results and Discussion

### 3.1. 3D Model Generation from 2D Medical Images

In a first step, we investigated the accuracy of the SSMs. Therefore, the SSMs were fit to 3D models generated from the 3D CT images in a leave-one-out cross-validation. The SSMs reach an average mean distance between fitted SSM and ground truth 3D model of approximately 0.2 mm. The average maximum distance between fitted SSM and ground truth 3D model was approximately 0.5 mm. In both cases, the average is taken over all bone SSMs and all CT images.

In the second step, we investigated the accuracy of the 3D shape reconstruction using the artificially generated data set. We present the results averaged over all finger bones and all 5 reference configurations. We reach an average Euclidean point-to-point surface distance norm of 0.03 mm, which correlates to an average Hausdorff norm of 1.3 mm. Adding the SSM fitting error to the reconstruction error, we still reach a high accuracy of approximately 1.5 mm mean surface distance. A box plot showing the results for each finger bone is presented in Figure 3. While the proposed method shows some outliers for the bones of the thumb, it performs very well on the other bones. For real data we can thus expect good performance and submilimetre accuracy.

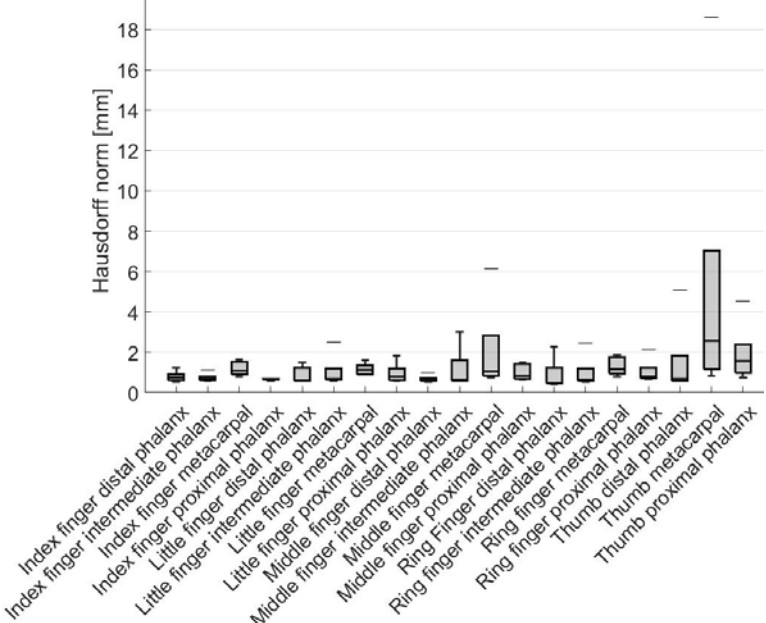

**Figure 3.** Statistics of Hausdorff distance [mm] for each finger bone summarized for all five reference configurations.

### 3.2. Individual Implant Design

Creation of Implant Design

During the design process of the implants, two additional factors are taken into consideration. First, the implant's head is equipped with an outer shell of full material, while the shaft is fortified with a solid core, thus ensuring the implant's basic stiffness. Second, a lightweight porous structure is incorporated in the implant's head and on the outer surface of the shaft to facilitate bone ingrowth and enhance the implant-bone interface. The porous domain consists of unit cells of TPMS structures, arranged in specified areas of the implant.

TPMS structures are mathematically smooth without sharp corners or edges, and can usually be described by one implicit equation. The mathematical expression of the Split-TPMS that is used for the porous structure outside of the implant is given by:

$$
\begin{aligned}
&+1.1(\sin(2x)\sin(z)\cos(y) + \sin(2y)\sin(x)\cos(z) \\
&\qquad + \sin(2z)\sin(y)\cos(x)) \\
&-0.2(\cos(2x)\cos(2y) + \cos(2y)\cos(2z) \\
&\qquad + \cos(2z)\cos(2x)) \\
&-0.4(\cos(2x) + \cos(2y) + \cos(2z))
\end{aligned}
\tag{9}
$$

The implicit modelling allows for specific adjustments, such as stretching the unit cells in order to attain desired properties and better alignment with the solid core of the implant geometry. Once the TPMS has been mathematically defined, a three-dimensional volume is generated by thickening the surface in both normal directions, followed by merging it with the solid components of the implant. The unit cell of the Split P-TPMS structure is illustrated in Figure 4.

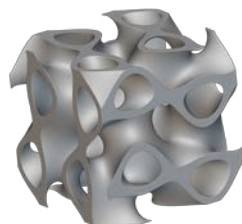

**Figure 4.** Unit cell of the Split P-TPMS.

For the finite element analysis, the resulting and oriented implicit implant geometries are converted into computer-aided-design (CAD) bodies and subsequently exported from nTopology. The entire workflow is illustrated in Figure 5.

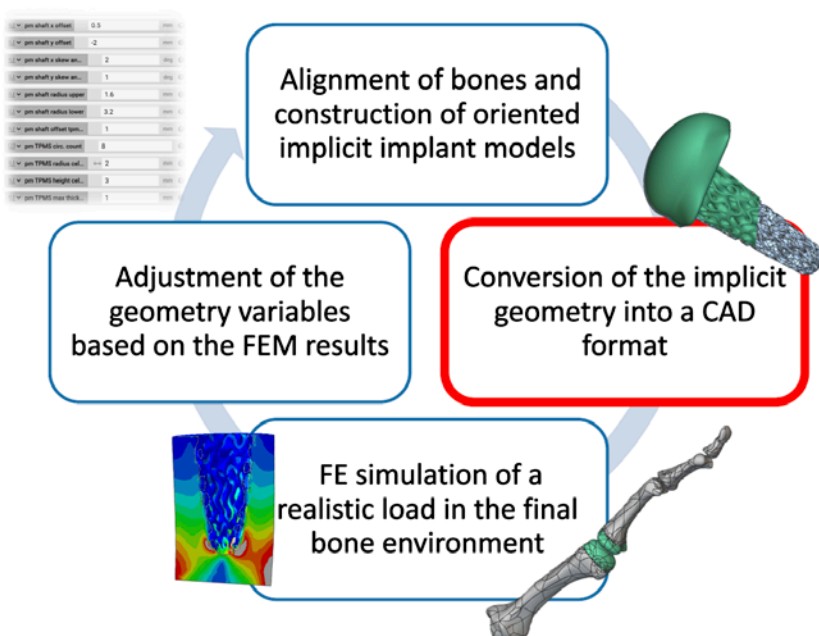

**Figure 5.** Process for implant design and data processing.

### 3.3. Additive Manufacturing of Silicon Nitride Specimen

The first $Si_3N_4$ test samples created by additive manufacturing via vat photopolymerization (CerAM VPP) were discs for biaxial bending test via ball-on-three-balls method. Afterwards, cylindrical SplitP-type TPMS components with a base plate were successfully

generated via 3D-printing followed by the manufacturing of a first complex finger implant also based on a SplitP design. The various test samples are presented in the following in sintered state (Figure 6).

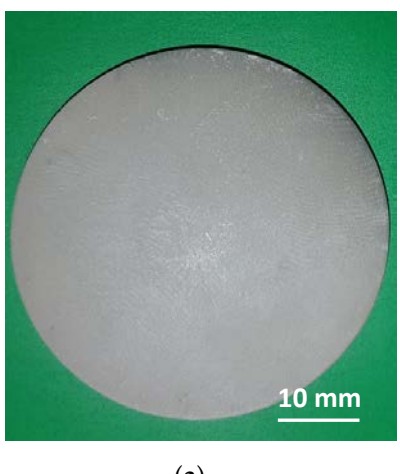 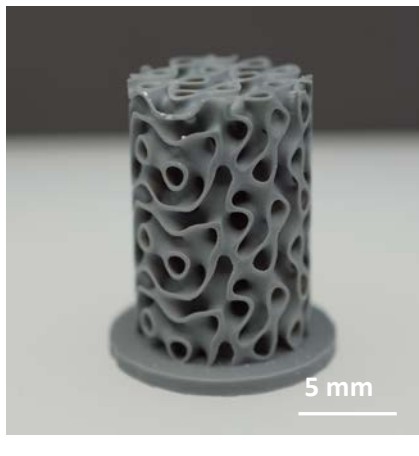 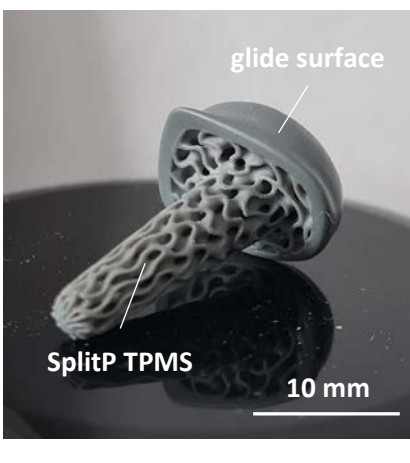

(**a**)                                                     (**b**)                                                     (**c**)

**Figure 6.** Sintered part of silicon nitride manufactured via CerAM VPP: (**a**) discs for biaxial bending tests; (**b**) a cylindrical TPMS (SplitP design) component on base-plate; (**c**) example of on finger implant with inner dense core, outer porous SplitP structure, and a polished glide surface.

The visual inspection reveals no obvious defects or cracks, and the resolution of the sintered components is of good quality in comparison to the nominal dimension.

### 3.4. Fracture Test of Silicon Nitride Specimen

3.4.1. Discs for Ball-On-Three-Balls Test in Comparison to 4-Point Bending Results

Three different $Si_3N_4$ ceramics were tested in B3B or standardized 4-point bending test geometry. VPP (via vat photopolymerization) $S_3N_4$ is the ceramic optimized for the additive manufacturing process via photopolymerization of a binder suspension; CIP (cold isostatic pressed) $S_3N_4$ is the ceramic based on the same $Si_3N_4$ granules as VPP with 6 wt% alumina and 5 wt% yttria [24]; and $Si_3N_4$-E10 SC [49] is a high temperature resistant benchmark that was gas pressure sintered at 1800 °C under 50 bar nitrogen atmosphere based on the same $Si_3N_4$ powder but 1.5 wt% alumina and 8.1 wt% yttria.

The B3B case test produces biaxial critical stresses in the centre of a disc, avoids edge influences, and is mechanically related to the double ring test. It produces a small effectively loaded volume and is very surface sensitive. Börger et al. [46] has validated the use of B3B in comparison to 4-point bending. He did a comparison using alumina with "ground" and "unsintered" surfaces. The application of effective volume for stress comparison was validated for this system. Danzer et al. [43] have shown that a surface finish required by ENV 843-1 is applicable for materials that have an effective volume $\geq 0.01$ mm$^3$ in the test, and thus silicon nitride. Two steps are required in the application: to apply surface sensitive testing with the B3B method for surface dominated structures, and to compare test results from the B3B method and 4-point bending. The methodical verification of the comparison of the B3B test and 4-point bending was shown by Danzer. An applicability of the distribution assumption for critical errors is checked via a Weibull plot (Figure 7) of the individual strength results. A high applicability was proven for the materials and test types since a high linear correlation was found. In detail, minor deviations from linearity are visible, representing high and low strength cases. The Weibull parameter m can be obtained from the slope in this diagram, and is high in all cases.

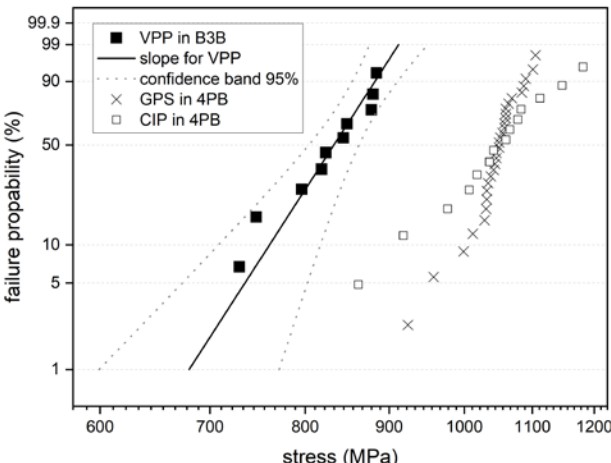

**Figure 7.** Weibull diagram for via vat photopolymerization manufactured (VPP) and cold isostatic pressed (CIP) binder based Si3N4 (adapted and redrawn from Ref. [24] 2022 Elsevier Ltd.), in comparison to gas pressure sintered (GPS) high strength Si3N4 (adapted and redrawn from Ref. [49] 2020 Elsevier Ltd.).

The comparison of VPP $Si_3N_4$ and $Si_3N_4$-E10 have revealed that the evolution of cyclic fatigue is similar (Figure 8), and therefore a similar resistance against crack growth is obtained. This can be concluded by the very parallel slope of strength reduction at increasing cycles. But VPP $Si_3N_4$ shows a broader distribution of underlying critical defects due to the additive process, and obviously has larger defects in comparison to the optimized $Si_3N_4$ E10 SC.

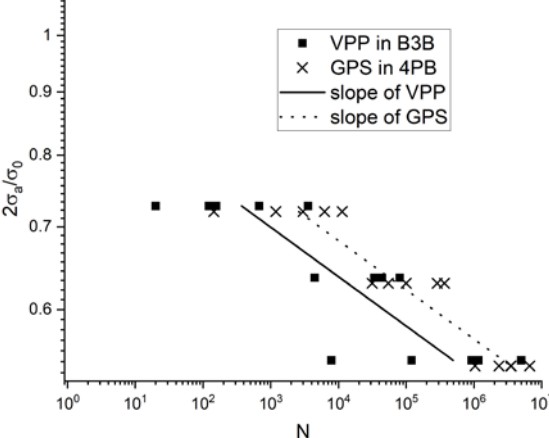

**Figure 8.** Cyclic fatigue-adjusted S-N data and normalized stress amplitude to critical stress for via vat photopolymerization manufactured (VPP) binder based Si3N4 (adapted and redrawn from Ref. [24] 2022 Elsevier Ltd.), in comparison to gas pressure sintered (GPS) high strength Si3N4 (adapted and redrawn from Ref. [49] 2020 Elsevier Ltd.).

Likewise, the resistance to cyclic cracks is lower for VPP $Si_3N_4$, which possibly could be the result of these larger defects, but the results are promising for the use in an innovative structured design.

3.4.2. Mechanical Behavior of Cylinder with SplitP TPMS

The cylindrical specimens were made with or without a mounting plate so that its influence on the uniaxial test of a porous specimen could also be tested (Figure 9). In addition, this build-up plate was tested on the side with spherical bearing (top) or on the flat opposite side (bottom).

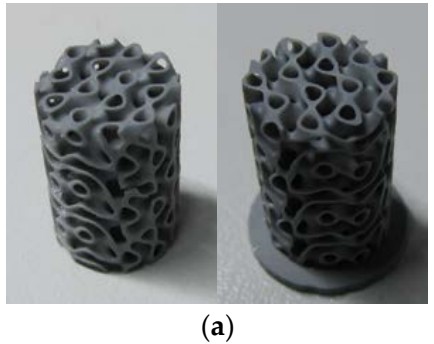
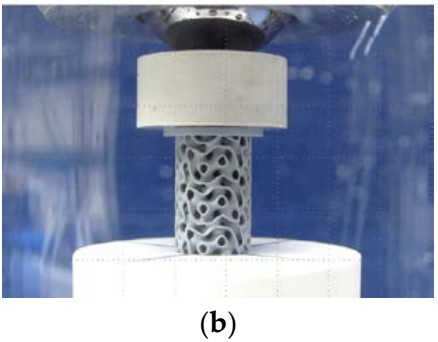

(**a**) (**b**)

**Figure 9.** Uniaxial compression of a Si$_3$N$_4$-cylinder with SplitP TPMS structure: (**a**) manufactured specimen with and without loading plate; (**b**) specimen inside the uniaxial test setup.

For the classification of the test in comparison with the later simulations, the recorded force-displacement curves were calculated into homogenized true stresses of the cross section and uniaxial mean strain by means of the dimensions (Figure 10). This means that the mechanical behavior can also be directly related to the surrounding bone material. Young's moduli of up to 17 GPa have been measured for bone material. Thus, it is possible to match the implant material with the highest modulus of elasticity (Si$_3$N$_4$ ceramics) to that of the bone by structuring. The achieved strength of this structure of 90 MPa is sufficiently high in comparison with biological material. To what extent the calculated loads can be compared with the actual physiological loads cannot yet be answered. For these reasons, a comparison will be sought via load simulations of each structure, shape, implant and loading scenario.

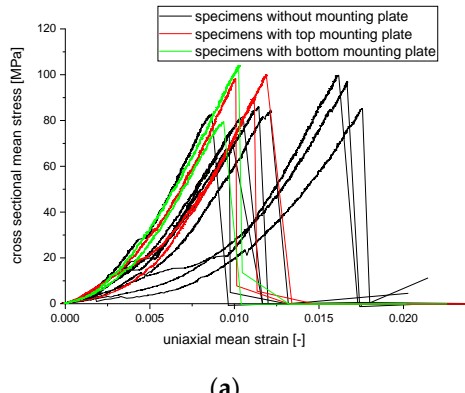
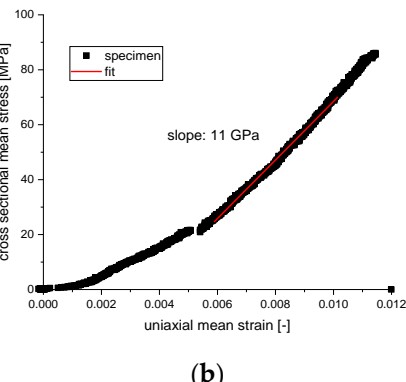

(**a**) (**b**)

**Figure 10.** Results of uniaxial compression of a Si$_3$N$_4$-cylinder with SplitP TPMS structure: (**a**) homogenized true stresses of the cross section and uniaxial mean strain by means of the dimensions; (**b**) fitted elasticity to stress-strain results.

### 3.5. Finite-Element-Modeling and Reliability Calculations

#### 3.5.1. Discs for Ball-on-Three-Balls Test and Calculation of Weibull-Parameter

The calculation of stress in the B3B test was already provided by Börger et al. [45]. However, to calculate the effect of structures on surfaces, of elasticity gradients, of geometric deviations and, based on the load, the effective volume necessary for the reliability calculation, own simulation sets were built. Here, the loading by the spheres was represented by rigid surfaces since the contact stresses formed are far from the biaxial stresses to be considered (Figure 11). Using the experimentally determined Weibull modulus $m = 18$, the corresponding effective volume was calculated to be $V_{eff,B3B} = 0.02$ mm$^3$, and using the characteristic fracture stress $\sigma_{0,B3B°} = 847$ MPa for the B3B test, the reference volume based parameters $V_0 = 1$ mm$^3$ und $\sigma_{0,B3B°} = 681$ MPa can now be calculated.

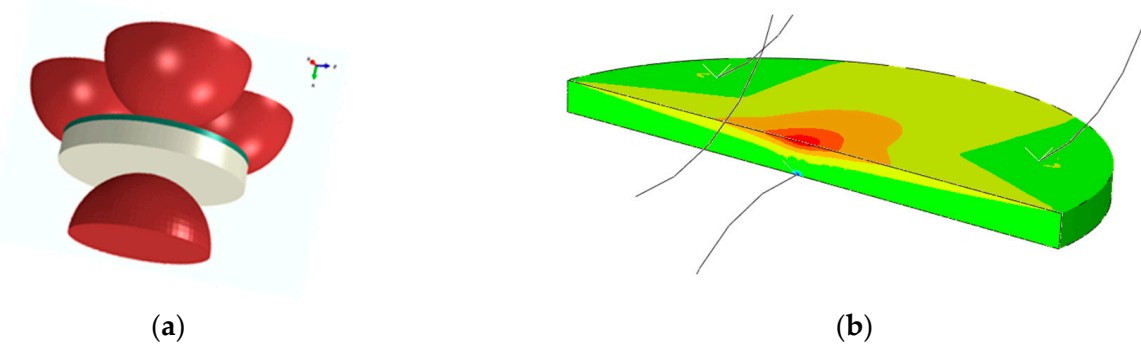

(**a**) (**b**)

**Figure 11.** Ball-on-three-balls test: (**a**) test geometry for cylindric plates with and without top layer; (**b**) maximal principal stress for cut view of loaded cylinder.

Since the internal load in the B3B test is not linearly coupled with geometric parameters, the effective volume must be provided not only as a function of the Weibull modulus *m*, but also as a function of the generated stress $F/t^2$ for the specific geometry (Figure 12).

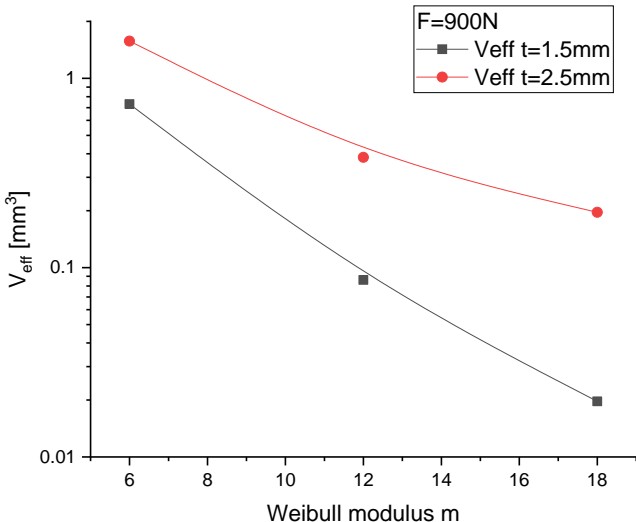

**Figure 12.** Ball-on-three-balls test: dependence of effective volume on Weibull module and loading parameter $F/t^2$.

### 3.5.2. Cylinder with Cylindrical SplitP TPMS in Comparison to Experimental Results

The creation of a high-quality mesh for the calculation of the effective volume is a compromise challenge between element fineness along the 3D curved surfaces and the limitation of the hardware. By simulation of the uniaxial compression test in full detail, the local stress ratio of the 1st principal stress to the uniaxial cross-sectional stress of 6.4 (stress ratio) can be obtained. For this geometry, an effective volume $V_{eff,cylinder}$ = 11.4 mm$^3$ and a characteristic stress $\sigma_{0,\,cylinder°}$ = 595 MPa in the structure valid for the geometry can be calculated (Figure 13). The strength parameters determined from the experiments of the B3B test for the case with the assumed optimised load application through a mounting plate (top) could be almost identical in value as well as a converted standard deviation.

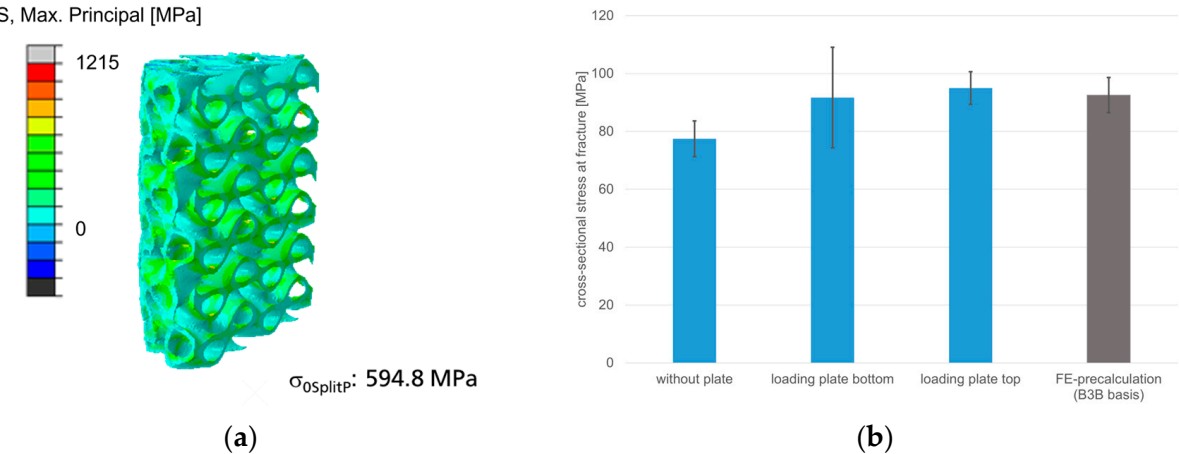

(**a**) (**b**)

**Figure 13.** Uniaxial compression test of SplitP TPMS structure: (**a**) maximal principal stress at 8 kN; (**b**) the comparison of calculated fracture stress in comparison to the experimental results.

### 3.5.3. Calculation of Implant Load with Gradient SplitP TPMS for a Diagonal Joint Load

There is no definite load limit defined for the implants positioned in the finger bones. By estimating a high mechanical compressive load for the joint only, a finger posture like that of a piano player was assumed with an angled finger posture and a load of 100 N. A maximum load applied to the joint was estimated at 45 °C for an abstracted preliminary study. For a first classification, a homogeneous elasticity of 10 GPa was assumed for the bone material in a simplified way (Figure 14). For this extremely localized implant load, an effective volume $V_{eff,implant} = 1.23 \times 10^{-4}$ mm$^3$ and a characteristic stress $\sigma_{0,implant45°} = 1123$ MPa are calculated. The used subroutine creates an output field parameter probability of survival for each element.

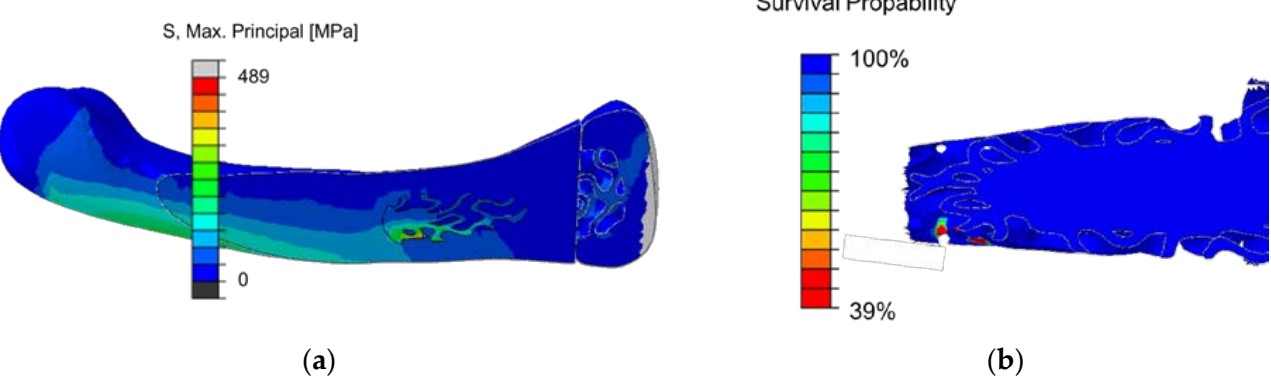

(**a**) (**b**)

**Figure 14.** Index finger implant in proximal bone: (**a**) maximal principal stress when implant is loaded with 45° to bone axis with 2.5 times of 100N reference load; (**b**) probability of survival for each finite element of a defined region.

The generated load in the implant is distributed in the form of a bending load deep into the implantation and is essentially determined in the local stress increase by the geometric shape of the TPMS SplitP structure following the shape of the implant (Figure 15). The evaluation and optimisation of such a solution presents itself as an ideal field of activity for the use of an AI on a design sequence. However, the load on the bone material should also be medically evaluated. For the simplified assumption of homogeneous elasticity, a moderate load of ~20 MPa results. Whether this load is super- or subcritical cannot be assessed at this stage.

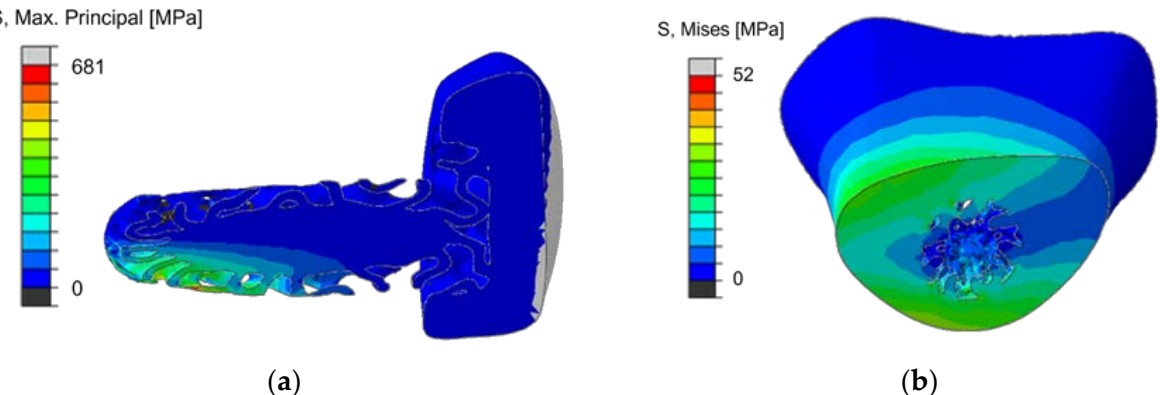

(**a**) (**b**)

**Figure 15.** Index finger implant in proximal bone: (**a**) cut view of maximal principal stress in the implant; (**b**) Cut view of von Mises equivalent stress in the bone.

However, first steps can be taken and the change of the load in the implant can be calculated via variation of the bone elasticity. Implant stress and bone elasticity are not linearly coupled, but the load increases significantly with decreasing bone modulus, according to the increasing proportion of the load balance under bending (Figure 16).

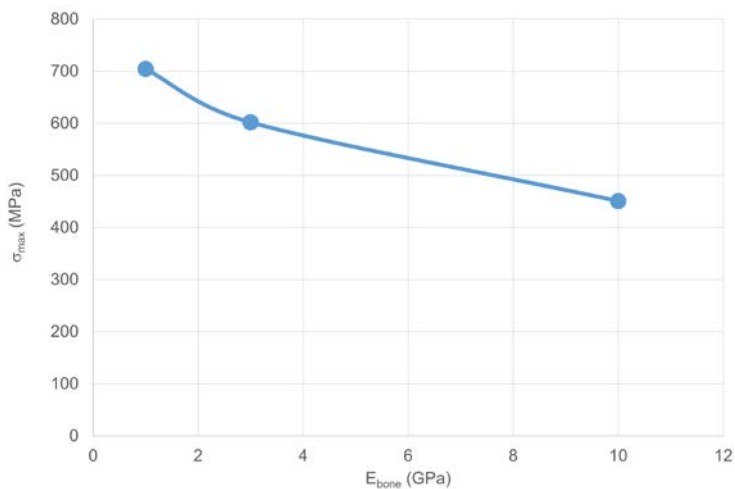

**Figure 16.** Dependence of implant loading stress on bone elasticity.

## 4. Conclusions

The concept of being able to manufacture precisely-fitting finger joint implants is already well developed. An automated process chain has been started from which individualised implant shapes made of any material, metallic or ceramic, can be produced quickly and safely in the future The implants that can be manufactured are still demonstrators. Nevertheless, the process chain up to the individual certified implant has been prepared or researched. The creation of such implant geometry is still dependent on further developments and technological accelerations in manufacturing and evaluation. AI-assisted software has proven to be a powerful tool.

Further technological developments and interface capabilities for automated evaluation of complex, individualized, structured implants are needed but underway.

For silicon nitride ceramics, a material and process development for the additive manufacturing process could be advanced to such an extent that a calculation of the reliability under load is possible since a suitable mechanical characterization with high surface sensitivity for the filigree structural elements could be achieved for the brittle material. The calculated value of a failure probability can be used as an evaluation parameter for brittle

ceramics in the targeted process chain. The integration into an AI-supported process chain prepared in this way is being worked out.

The main impacts and challenges of this study were:

- The ceramics created by VPP can be reliably applied to filigree structures.
- The TPMS structures of the implant can be created in a graded form along the curvature of the complex implant. A full workflow for a specific gradient generation of a TPMS to solid structure was achieved in a CAD-nTopology loop for individual implants and bones.
- SplitP TPMS structures have been validated for brittle materials as excellent elasticity-mitigating structures (3.6%) with low stress factor (6.4).
- The ball-on-three-ball test is predestined for the brittle materials of submilimetre VPP ceramic structure.
- A full workflow converts joint bone models, matches and aligns them to implants.
- The submilimetre accuracy of the AI-based 3D shape construction of 2D real data is expected to be good, as it was validated on artificial reconstruction loop 3Dto2Dto3D.

The workflow shown here is an example of one of many different ways and methods to address such an issue in order to create reliable individualized implants. We have stated the interface problems are solved and demonstrated the mechanical reliability of a matching of characterization and testing. However, further interfaces remain to achieve for a closed process from automated or artificial intelligence-driven sequences. In the future, a development by Fraunhofer Research could help fingers with destroyed or damaged joints regain their mobility by using a concept in which precisely fitting finger joint implants can be created: individualised implants made of metallic or ceramic materials produced quickly, safely and certified in an automated process chain.

**Author Contributions:** Conceptualization, C.K., Y.M.L., A.S., J.G., E.Z., F.G., S.Y.-S.; writing and investigation, C.K., E.S.-F., A.R., M.C., Y.M.L.; review and editing, A.K., E.Z., A.S., J.G.; resources of anonymized patient data, F.G., S.Y.-S. All authors have read and agreed to the published version of the manuscript.

**Funding:** This research received no external funding. The project FingerKit is an internal project of the Fraunhofer-Gesellschaft zur Förderung der angewandten Forschung e.V.

**Institutional Review Board Statement:** Ethical review and approval were waived for this study due to the fact that the anonymized patient data used to build the statistical shape models can no longer be mapped to anyone.

**Informed Consent Statement:** Patient consent was waived due to the fact that the anonymized patient data used to build the statistical shape models can no longer be mapped to anyone.

**Data Availability Statement:** Not applicable.

**Acknowledgments:** We would like to thank Nathanael Girges (from Fraunhofer IAPT) for his support in adapting the general implant geometry to individual, patient-specific implant geometries.

**Conflicts of Interest:** The authors declare no conflict of interest.

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
