# Peer review of "Design of Reliable Remobilisation Finger Implants with Geometry Elements of a Triple Periodic Minimal Surface Structure via Additive Manufacturing of Silicon Nitride"

_2571-8800, doi:10.3390/j6010014_

Round 1

Reviewer 1 Report

In this article, finger joint implants with three-cycle minimal surface structure were fabricated by additive manufacturing technique using silicon nitride based ceramics. It also analyzes the load and characteristic strength matching relationship by conducting experimental tests on model specimens to generate and match failure probability estimates. The feasibility of silicon nitride based ceramic structures fabricated in this manner was confirmed using this approach. The article has some problems that need to be improved.

1, Please explain how the coupled motion between joints is considered after implantation of the printed implant.

2, What is the TPMS structure, which is not explained in detail in the article?

3, The specific parameters for the implicit function design and simulation of the model are not listed.

4, How do the results of this model design study compare to others in the literature should be added to the article?

Author Response

In this article, finger joint implants with three-cycle minimal surface structure were fabricated by additive manufacturing technique using silicon nitride based ceramics. It also analyzes the load and characteristic strength matching relationship by conducting experimental tests on model specimens to generate and match failure probability estimates. The feasibility of silicon nitride based ceramic structures fabricated in this manner was confirmed using this approach. The article has some problems that need to be improved.

We have revised the manuscript using the excellent propositions and questions of the reviewers. A point to point answer was given to the detailed aspects. More references were used that help the understanding and positioning of our paper to the multidisciplinary challenge of problem solving.

1, Please explain how the coupled motion between joints is considered after implantation of the printed implant. -->

Thank you for your valuable feedback. Regarding the movement of the joint, we have added a paragraph and a picture showing the alignment of the joints under point 2.2.

2, What is the TPMS structure, which is not explained in detail in the article?

We have added new text to paragraph 3.2.1 so that it now gives a general explanation of TPMS structures as well as the specific application of these in the project. In addition, Figure 4 shows one of the unit cells mentioned in the section above it.

3, The specific parameters for the implicit function design and simulation of the model are not listed.

Good point. The mathematical description and the adjustments made to the unit cells were added.

4, How do the results of this model design study compare to others in the literature should be added to the article?

Truly, this aspect was an open point in our multidisciplinary storyline, and we have tried to explain the aim of our paper added under 1.2. Materials and processes for implant manufacturing.

Reviewer 2 Report

The authors propose a methodology, AI based, for the design of silicon nitride finger implants. The 3D model after FEM validation, and material characterization has been 3D printed. The TPMS structure used for the realization of the finger implant has been tested as well.

Silicon nitride ceramics have been applied fairly recently as medical implant, they show promising properties such as phase stability, mechanical resistance, bacteriostaticity, but the road to the perfect biomaterial is still long and studies like this one, that connect the material to a design methodology strictly correlated to the application are more than indispensable to optimize both the material and manufacturing process.

The paper seems, in some parts roughly written, the hypothesis, the scope, the results, discussion and the conclusions are not always clear, but the applied methodologies sound correct. The manuscript has few weaknesses that need clarification and adjustment. In order to reach the standards of the journal this manuscript require major revisions, carefully addressing the following points.

1.       Title: I suggest to avoid uncommon terms and acronyms in the title, as “additive cerAMfacturing”. “Additive Manufacturing of ceramics” or a similar terminology would be preferred, nevertheless, clearer also for non-specialists.

2.       English language must be deeply revised. Some sentences do not sound correct (e.g. lines 369-370, 470-471), but in other part of the text the meaning is far from clear (e.g. 482-484).

3.       Abstract, lines 23-25: the phase “AI-supported software can calculate three-dimensional models of the finger bones from two-dimensional X-ray images and correct a potential malposition of the fingers” seems quite speculative considering the state-of-the-art. The training of the AI, on single/double 2D images, in this research is still in the working progress stage. I propose to change the sentence suggesting the AI-driven design, without proper 3D CT images, as a close-future perspective.

4.       The Introduction paragraph is missing of important references related to Si3N4 applied as a biocompatible material and general characteristics of the material, but also examples of AI application in the medical field, or at least for bone regeneration/artificial bone design.

5.       Lines from 65 to 69: Authors affiliation are already present in the proper section. It’s not necessary to report again in the text the institutes that took part in the research, thus I suggest to remove those lines.

6.       Line 70: The funding project, “fingerkit”, that sustained this research should be reported in the proper section (page 9 line 531: Funding: … - which is empty) and not in the text.

7.       Lines 87-89: “In the same manner the fundamental work of the Fraunhofer-Institute for Toxicology and Experimental Medicine ITEM will not be part of shown results. Essential work on certification and osseointegration is in progress.” there is no need to report things that are not present in the text, since they do not add anything to the reader. If the authors think that their in-progress work is needed, I suggest adding, at some point in the text, a perspective paragraph; they can also acknowledge in the proper section (page 16, line 539) ITEM institute.

8.       Lines 108-109: an artificial (like silicon nitride) term of comparison for the elastic modulus is missing; also, a bracket is missing in that sentence.

9.       Lines 181, 186-189 and 211: MDPI does not clearly state how to refer to a software (Mevislab, Matlab, nTopology, etc.), but they might be added as a reference moving part of the text and website there, to simplify the reading.

10.   Line 234: TMPS acronym misspelled ad not yet defined, nor clearly described - directly or by citation - the structure generation (software, morphology,  etc.)

11.   Paragraph 2.4: description of the uniaxial test setup must be added.

12.   Figure 2: Y-axis name and unit are missing.

13.   Paragraph 3.4.1 line 363: “4-point bending test” must be described in materials and methods.

14.   Line 369: Si3N4-E10 SC sample must be described in materials and methods (powder, manufacturing procedure, sintering and machining); if already reported in another paper it must anyway reported in material, with a clear reference.

15.   Line 370: ”… for pas pressure processes” the meaning is not clear.

16.   B3B test is “suitable for as fire surfaces” (line 259) but also “very surface sensitive” (line 374). How can the authors guarantee that all the samples surface are similar since no surface machining is applied nor any profilometry analysis performed on the as-is samples?

17.   Figure 5 and Figure 6: in the case Si3N4 E10 SC data have been previously published, reference should be added also in the figure captions.

18.   Paragraph 3.4.2, lines from 416 to 419 must be rewritten, since the meaning is not clear. Which properties in 17GPa, and “an actual classification with possible physiological loads is initially only possible via the simulations carried out later”. Later in the text (below?) or later in future (to be carried out?). Moreover, the comparison between cylindrical SplitP TPMS compression and B3B test seems not well addressed in the paragraph, despite the title.

19.   Figure 10. I suggest to change X-axis name from “m” to “Weibull modulus”

20.   Line 452-454: “uniaxial cross-sectional stress of 6.4” How this result is obtained? Furthermore, the sentence contains repetitions.

21.   Lines 470-471. Please double check the text avoiding repetition - “For this implant load …. this implant load and ..”.

22.   I suggest changing paragraph “4. Discussion” in “Conclusions”. Moreover, Conclusion paragraph seems too discursive for a scientific paper. Numerical results and achieved goal must be emphasized more than future perspectives.

23.   Many typos are present overall the manuscript: capital letter missing in most of the figure’s captions, incorrect chemical formulas - Si3N4 instead of Si3N4 - (e.g. 366, 368, 388, etc.), missing references (line 232), punctuation (e.g. 109, 389, 499, etc.), markers in captions of Fig. 5 and 6 are not correct (rectangle instead of asterisk). Please carefully double check the entire paper, the list is not exhaustive.

Author Response

The authors propose a methodology, AI based, for the design of silicon nitride finger implants. The 3D model after FEM validation, and material characterization has been 3D printed. The TPMS structure used for the realization of the finger implant has been tested as well.

Silicon nitride ceramics have been applied fairly recently as medical implant, they show promising properties such as phase stability, mechanical resistance, bacteriostaticity, but the road to the perfect biomaterial is still long and studies like this one, that connect the material to a design methodology strictly correlated to the application are more than indispensable to optimize both the material and manufacturing process.

The paper seems, in some parts roughly written, the hypothesis, the scope, the results, discussion and the conclusions are not always clear, but the applied methodologies sound correct. The manuscript has few weaknesses that need clarification and adjustment. In order to reach the standards of the journal this manuscript require major revisions, carefully addressing the following points.

We have revised the manuscript using the excellent propositions and questions of the reviewers. A point to point answer was given to the detailed aspects. More references were used that help the understanding and positioning of our paper to the multidisciplinary challenge of problem solving.

  1. Title: I suggest to avoid uncommon terms and acronyms in the title, as “additive cerAMfacturing”. “Additive Manufacturing of ceramics” or a similar terminology would be preferred, nevertheless, clearer also for non-specialists.

We were used to have this terminology but since you judge this to be shortened, we should use this standard description instead.

  1. English language must be deeply revised. Some sentences do not sound correct (e.g. lines 369-370, 470-471), but in other part of the text the meaning is far from clear (e.g. 482-484).

We have revised these sentences.

  1. Abstract, lines 23-25: the phase “AI-supported software can calculate three-dimensional models of the finger bones from two-dimensional X-ray images and correct a potential malposition of the fingers” seems quite speculative considering the state-of-the-art. The training of the AI, on single/double 2D images, in this research is still in the working progress stage. I propose to change the sentence suggesting the AI-driven design, without proper 3D CT images, as a close-future perspective.

We like to follow your suggestion to state that AI-supported software was trained to do so. That is an important point.

  1. The Introduction paragraph is missing of important references related to Si3N4 applied as a biocompatible material and general characteristics of the material, but also examples of AI application in the medical field, or at least for bone regeneration/artificial bone design.

We have added some paragraphs on Si3N4 as a bioceramic, on AI application in the medical field, and the challenge of bone design. Thank you for the advice.

  1. Lines from 65 to 69: Authors affiliation are already present in the proper section. It’s not necessary to report again in the text the institutes that took part in the research, thus I suggest to remove those lines.

We have solved this issue together with suggestion 7. Thank you.

  1. Line 70: The funding project, “fingerkit”, that sustained this research should be reported in the proper section (page 9 line 531: Funding: … - which is empty) and not in the text.

We have included, that this is an internal project of Fraunhofer Gesellschaft E.V. as an additional information under funding.

  1. Lines 87-89: “In the same manner the fundamental work of the Fraunhofer-Institute for Toxicology and Experimental Medicine ITEM will not be part of shown results. Essential work on certification and osseointegration is in progress.” there is no need to report things that are not present in the text, since they do not add anything to the reader. If the authors think that their in-progress work is needed, I suggest adding, at some point in the text, a perspective paragraph; they can also acknowledge in the proper section (page 16, line 539) ITEM institute.

Thank you for this suggestion. We have moved this part partially to the end or left out.

  1. Lines 108-109: an artificial (like silicon nitride) term of comparison for the elastic modulus is missing; also, a bracket is missing in that sentence.

We have added youngs moduli and this should really help the reader. Thank you.

  1. Lines 181, 186-189 and 211: MDPI does not clearly state how to refer to a software (Mevislab, Matlab, nTopology, etc.), but they might be added as a reference moving part of the text and website there, to simplify the reading.

Good suggestion. We have done that.

  1. Line 234: TMPS acronym misspelled ad not yet defined, nor clearly described - directly or by citation - the structure generation (software, morphology, etc.)

We have rephrased the text in paragraph 3.2.1 and extended it so that it now gives a general explanation of TPMS structures as well as the specific application of these in the project. In addition, Figure 4 shows one of the unit cells mentioned in the section above it.

  1. Paragraph 2.4: description of the uniaxial test setup must be added.

added

  1. Figure 2: Y-axis name and unit are missing.

added

  1. Paragraph 3.4.1 line 363: “4-point bending test” must be described in materials and methods.

added

  1. Line 369: Si3N4-E10 SC sample must be described in materials and methods (powder, manufacturing procedure, sintering and machining); if already reported in another paper it must anyway reported in material, with a clear reference.

The reference to this material was given at the end of the sentence, which was misleading and corrected.

  1. Line 370: ”… for pas pressure processes” the meaning is not clear.

Correction was done: “high temperature resistant benchmark that was gas pressure sintered at 1800 °C under 50 bar nitrogen atmosphere based on the same Si3N4 powder”

  1. B3B test is “suitable for as fire surfaces” (line 259) but also “very surface sensitive” (line 374). How can the authors guarantee that all the samples surface are similar since no surface machining is applied nor any profilometry analysis performed on the as-is samples?

The applicability for different surface conditions was tested by Börger and Danzer et al. References are now included.

  1. Figure 5 and Figure 6: in the case Si3N4 E10 SC data have been previously published, reference should be added also in the figure captions.

added

  1. Paragraph 3.4.2, lines from 416 to 419 must be rewritten, since the meaning is not clear. Which properties in 17GPa, and “an actual classification with possible physiological loads is initially only possible via the simulations carried out later”. Later in the text (below?) or later in future (to be carried out?).

We have rewritten this paragraph.

Moreover, the comparison between cylindrical SplitP TPMS compression and B3B test seems not well addressed in the paragraph, despite the title.

The title has been revised, since the comparison has to be done using simulations.

  1. Figure 10. I suggest to change X-axis name from “m” to “Weibull modulus”

Yes

  1. Line 452-454: “uniaxial cross-sectional stress of 6.4” How this result is obtained? Furthermore, the sentence contains repetitions.

We have rewritten this paragraph.

  1. Lines 470-471. Please double check the text avoiding repetition - “For this implant load …. this implant load and ..”.

Yes, we have rewritten these sentences.

  1. I suggest changing paragraph “4. Discussion” in “Conclusions”.

We have followed this suggestion.

Moreover, Conclusion paragraph seems too discursive for a scientific paper. Numerical results and achieved goal must be emphasized more than future perspectives.

This aspect is very relevant. In fact, the paper in this form is a showcase of a way of different ways and methods to solve such a question. We have stated the interface problems solved and demonstrated the mechanical reliability of a matching of matched characterization and testing. However, further interfaces remain to achieve a closed process from automated or artifial intelligence driven sequences.

  1. Many typos are present overall the manuscript: capital letter missing in most of the figure’s captions, incorrect chemical formulas - Si3N4 instead of Si3N4 - (e.g. 366, 368, 388, etc.), missing references (line 232), punctuation (e.g. 109, 389, 499, etc.), markers in captions of Fig. 5 and 6 are not correct (rectangle instead of asterisk). Please carefully double check the entire paper, the list is not exhaustive.

We have tried and hopefully found everything. Thank you.

Round 2

Reviewer 1 Report

there are no more questions.

Reviewer 2 Report

The paper, in this form, contain plagiarism. I suggest the editor to reject it.

Sentences have been copied as is, infringing the Ethics Guidelines related to Authorship and Plagiarism of MDPI.

some example:

line 182-183 "Additive manufacturing technologies have unlocked new possibilities for bone tissue engineering" is taken from the uncited source https://patents.google.com/patent/WO2020033607A1/en

line 183-185 "Long-term regeneration of normal anatomic structure, shape, and function is clinically important subsequent to bone trauma, tumor, infection, nonunion after fracture, or congenital abnormality. Due to the great complexity in structure and properties of bone across the population, along with variation in the type of injury or defect, currently available treatments for larger bone defects that support load often fail in replicating the anatomic shape and structure of the lost bone tissue" comes from the cited source [26] doi: 10.1089/3dp.2015.0001

line 189-190 "Triply periodic minimal surfaces (TPMS) are a non-intersecting 3D surface characterized by a zero value of mean curvature at each point" belongs to uncited source doi:10.1016/j.matdes.2021.110074

"different from classical lattice or foam structures, the smooth surfaces of TPMS are suitable for cells to attach and grow. Moreover, the high-volume specific surface areas and the highly interconnected porous architectures can supply enough space for the transport of nutrition and waste. TPMS structures have been successfully applied in current biological engineering. The compression performances have been discussed with different manufacturing materials. More studies of heterogeneous TPMS and multiscale TPMS are needed." is copied from different parts of the cited source [27] doi:10.1088/2631-7990/ac5be6

lines 464-469 "confirms that charging with effective volume is useful for tensions comparison. Danzer et al. [42] haves shown that a surface finish made according to ENV 843–1 can be applied well with an eff. volume ≥ 0,01 mm3 for silicon nitride. Since the method is supposed to provide results for filigree manufactured structures made by AM, the comparison with the B3B method is interesting to find out difference. The methodological verification of the comparison of B3B and 4-Point bending has been shown." comes from cited source [24] doi:10.1016/j.jeurceramsoc.2022.10.011